# MetaRibo-Seq measures translation in microbiomes

Brayon J. Fremin[1], Hila Sberro[1,2] & Ami S. Bhatt [1,2✉]

No method exists to measure large-scale translation of genes in uncultured organisms in microbiomes. To overcome this limitation, we develop MetaRibo-Seq, a method for simultaneous ribosome profiling of tens to hundreds of organisms in microbiome samples. MetaRibo-Seq was benchmarked against gold-standard Ribo-Seq in a mock microbial community and applied to five different human fecal samples. Unlike RNA-Seq, Ribo-Seq signal of a predicted gene suggests it encodes a translated protein. We demonstrate two applications of this technique: First, MetaRibo-Seq identifies small genes, whose identification until now has been challenging. For example, MetaRibo-Seq identifies 2,091 translated, previously unannotated small protein families from five fecal samples, more than doubling the number of small proteins predicted to exist in this niche. Second, the combined application of RNA-Seq and MetaRibo-Seq identifies differences in the translation of transcripts. In summary, MetaRibo-Seq enables comprehensive translational profiling in microbiomes and identifies previously unannotated small proteins.

[1] Department of Genetics, Stanford University, Stanford, CA 94305, USA. [2] Department of Medicine (Hematology & Blood and Marrow Transplantation), Stanford University, Stanford, CA 94305, USA. ✉email: asbhatt@stanford.edu

The organisms within fecal microbiomes likely have myriad biological functions, most of which are unknown. To date, methods have excelled at describing the taxonomic composition of such communities; however, assigning and defining functions of the individual organisms or communities of bacteria has been challenging[1]. An ideal method to study biological functions within a complex community would allow simultaneous enumeration of all of the proteins, lipids, and other macromolecules within the mixture. Unfortunately, this is not feasible with current technologies. For example, at present, only a small subset of proteins present within a microbiome sample ($\sim 10^2 - 10^4$) can be simultaneously quantified with metaproteomics[2]; thus, this presents a major challenge in attempting to obtain accurate measurements of the full array of the estimated $10^7 - 10^8$ bacterial proteins that likely exist in human fecal samples[3]. It is especially challenging to detect small proteins as the likelihood of detection of a protein is directly correlated to its length; because of this, small proteins are less likely to be detected than large proteins[4]. Thus, current proteomic methods lack the dynamic range required to comprehensively study the human fecal microbiota proteome[5].

Given that proteomics is limited in both throughput and yield, and it is also dependent on accurate databases of protein sequences, some have focused on enumerating the gene content of a community to determine its potential functions[6]. This has been further enabled by improved gene annotation in metagenomes, to include previously overlooked genes, such as those that are short in length (<150 nucleotides)[4]. Despite this progress, the presence of a predicted gene or even its RNA transcript does not confirm that the predicted gene encodes a protein.

In contrast to transcriptomic profiling, Ribo-Seq is a method that quantifies protein synthesis[7,8]. Promisingly, Ribo-Seq generally correlates more strongly to protein abundance than transcriptomics in eukaryotes[9–11]; however, this correlation has not yet been described in bacteria. Furthermore, most bacterial ribosome profiling studies published to date have been performed in model organisms such as *Escherichia coli* and *Bacillus subtilis*. These studies have been enabled by adapting eukaryotic Ribo-Seq protocols with modifications such as using chloramphenicol to inhibit translation and micrococcal nuclease (MNase) to enrich for ribosome footprints[11–15]; these methodological modifications enable a relatively high-throughput snapshot of bacterial translation[9]. While powerful, these studies have a major limitation— nearly all studies of protein synthesis in bacteria have been restricted to pure, liters-range cultures (requiring up to milligram-level RNA input). This limitation has resulted in a barrier to studying translation in microbiomes or in culture-free contexts, and is the result of several methodological challenges including low extraction yield, low purity, and the lack of informatic frameworks to study organisms without reference genomes.

In this work, we overcome many of these limitations and report a method that allows for simultaneous ribosome profiling in microbiomes without the need for a large-scale, purified cultures. We benchmark the performance of MetaRibo-Seq against other technologies using mock communities, apply the method to several human fecal samples, and report the utility of this method in identifying small genes that were previously unannotated.

## Results

### The MetaRibo-Seq workflow
MetaRibo-Seq is an experimental and computational approach that enables simultaneous, high-throughput ribosome profiling on a fecal mixture of microorganisms (Fig. 1a). We found that ribosome profiling can be performed on frozen fecal samples stored in RNAlater[16,17] (Ambion), an RNA-preserving solution. Unlike some existing protocols[14,18], our ribosome profiling protocol first introduced chloramphenicol during lysis. After lysis, we introduced an ethanol precipitation step; this step filtered out fecal debris and also concentrated RNA and any RNA-containing complexes such as ribosomes[19]. MNase treatment was then performed on a crude purification of nucleic acids and nucleic acid-bound complexes to degrade any unprotected DNA or RNA. MetaRibo-Seq used roughly an order of magnitude less RNA and MNase compared to traditional bacterial isolate Ribo-Seq protocols (see "Methods")[14,20]. For the computational part of the workflow, a major challenge was determining how to deal with short reads and poor or incomplete reference genomes, the latter of which is an inherent challenge of working with fecal samples. To overcome these challenges, we used a de novo approach to build reference genomes or genome fragments, annotate genes, and map reads to those references (see "Methods", Fig. 1b). Sequences detected by metatranscriptomics and MetaRibo-Seq were aligned to the assemblies to determine which genes were being transcribed or actively translated within the samples.

### MetaRibo-Seq of a mock bacterial community strongly correlates with standard Ribo-Seq
To benchmark the effectiveness of our approach, we performed multi-omics (metagenomics, metatranscriptomics, Ribo-Seq, MetaRibo-Seq, and proteomics) on a mock bacterial community of *E. coli*, *B. subtilis*, and *Staphylococcus aureus*. The complexity of the mock community was limited by the input requirement of traditional Ribo-Seq, which requires large volumes of cultured organisms; thus, while it would have been ideal to carry out the benchmarking experiment on a more diverse mock community, the limitations of Ribo-Seq preclude this, highlighting another strong need for a method such as MetaRibo-Seq. All RNA-Seq experiments performed in this work used an extended RNA fragmentation reaction followed by sequencing of the resulting small mRNA fragments; extended fragmentation is standard when performing both RNA-Seq and Ribo-Seq[7,21] as it controls for library preparation and size biases. We found that the taxonomic distribution of sequencing reads and detected peptides (Fig. 2a) were concordant between methods. As expected, we found that the Gram-positive bacteria within the mock mixture (*B. subtilis* and *S. aureus*) were relatively underrepresented compared to the Gram-negative bacterium *E. coli* for all technologies, suggesting that the standard lysis approaches in the field are likely biased to easily lysed Gram-negative bacteria. The bias was, however, consistent across sequencing technologies. We found that our standard proteomic protocol, which has a different standard lysis protocol[22] (including sodium dodecyl sulfate (SDS); see "Methods") than was used for the DNA and RNA sequencing protocols, resulted in the most even detection of proteins between taxa (Fig. 2a).

Ribo-Seq experiments classically show peak signal at the start codon of genes. MetaRibo-Seq signal exhibited this expected property with strong signal detected at predicted start codons (Fig. 2b). Ribo-Seq and MetaRibo-Seq were strongly correlated (Pearson's $r = 0.9$), suggesting that MetaRibo-Seq was comparable to the gold-standard approach, Ribo-Seq (Fig. 2c). In *E. coli*, we found similar signal distribution across the start of genes for Ribo-Seq and MetaRibo-Seq (Fig. 2d). In general, we found that metatranscriptomics, MetaRibo-Seq and Ribo-Seq signal all correlate to protein abundance as measured by metaproteomics (Supplementary Figs. 1–3). MetaRibo-Seq correlated more strongly to protein levels than metatranscriptomics in the aerobically grown mock community (Supplementary Fig. 1), as well as an anaerobic mock community (Supplementary Fig. 2) and low diversity fecal sample (Supplementary Fig. S3).

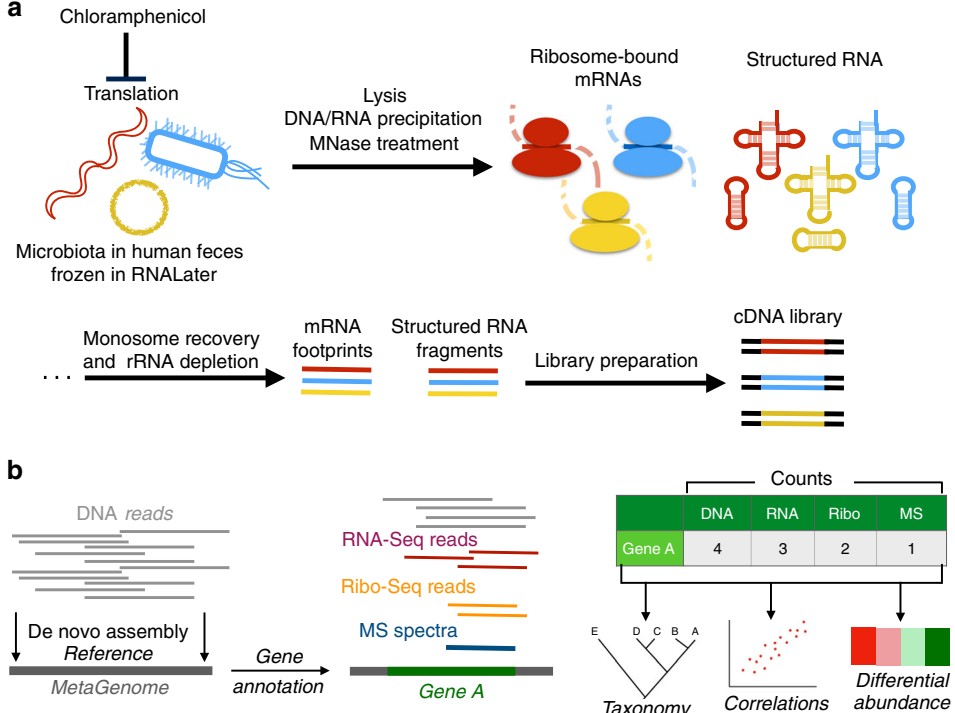

**Fig. 1 Workflow of ribosome profiling. a** Experimental workflow of MetaRibo-Seq. Chloramphenicol halts translation, bacteria within the community are lysed, MNase is used to create footprints, and footprints are converted to sequencing libraries. **b** Computational workflow of the multi-omics approach. De novo assemblies are created and annotated, predicted genes are quantified at multi-omic levels, and taxonomy, correlations, and differential abundance are determined from these results.

**MetaRibo-Seq enables simultaneous measurement of translation in fecal microbiome samples**. To evaluate the feasibility of performing MetaRibo-Seq on complex samples, such as human fecal samples, we performed MetaRibo-Seq, metagenomics and metatranscriptomics on four taxonomically diverse fecal samples from four different human subjects (A—healthy adult, B—patient with hematological disorder undergoing treatment, C—patient with cancer undergoing treatment, D—patient with Alzheimer's disease). We also performed multi-omics on a fifth, low diversity fecal sample from a patient with a hematological disorder who was undergoing antibiotic treatment with metronidazole—Sample E. The percentage of classified reads across samples and technologies varied, ranging from 33 to 91% (Supplementary Data 1). Taxonomic differences at the genus level existed between technologies across samples, with replicates displayed, where applicable (see "Methods", Fig. 3a). Replicates were strongly correlated, ranging from Pearson's $r$ of 0.86−0.93 (Supplementary Data 2).

We next tested if MetaRibo-Seq more strongly correlates to protein abundance than metatranscriptomics in these fecal samples. While we found that MetaRibo-Seq correlates more strongly to metaproteomics than metatranscriptomics in Sample E (Supplementary Fig. 3), MetaRibo-Seq did not reproducibly correlate more strongly to proteomics than metatranscriptomics in diverse samples (Supplementary Fig. 4). In diverse samples, metatranscriptomics and MetaRibo-Seq were significantly enriched ($p$ value $< 2^{-16}$) in signal for proteomically detected proteins (Supplementary Fig. 4). This suggests that metaproteomics on diverse samples is only capturing proteins that are highly abundant, as is expected, and thus does not sensitively recapitulate the diversity present in proteomes from highly diverse microbiome samples. In summary, metaproteomic limitations made it difficult to conclude if MetaRibo-Seq correlates more strongly to protein levels than metatranscriptomics in diverse fecal samples.

**MetaRibo-Seq of diverse fecal samples displays expected Ribo-Seq signal characteristics**. A standard test of Ribo-Seq signal validity is whether signal is locally enriched within coding regions and especially enriched for start and stop regions. As expected, we observed strong MetaRibo-Seq signal corresponding to predicted open reading frames (ORFs) with pronounced signal drop off outside of the start and stop codons for Samples A through D (Fig. 4a–d). Start and stop codons had the strongest signal (Fig. 4a–d). This signal distribution was not observed in RNA-Seq (Supplementary Fig. 5). We required perfect, unique matches of these ribosome footprints to de novo references to ensure proper mapping (see "Methods", Supplementary Data 3). Surprisingly, MetaRibo-Seq also displayed some weak signs of overall codon resolution (Supplementary Fig. 6), although this does appear to vary by taxon (Supplementary Fig. 7). Based purely on raw signal, these findings collectively suggest that MetaRibo-Seq is capturing ribosome-bound footprints, as expected.

**MetaRibo-Seq identifies gene-wide differences in translation efficiencies and regulation**. MetaRibo-Seq and metatranscriptomics can be used together to identify genes that are translated at significantly different levels than transcribed in fecal microbiomes. From the 866,832 genes encoded by the four metagenomes (samples A−D) analyzed, 42,267 (4.9 %) genes displayed transcriptional data that was significantly different than MetaRibo-Seq data (see "Methods", Supplementary Data 4, Supplementary Fig. 8). We clustered the 42,267 genes based on 70% amino acid similarity to create 32,277 clusters of homologs (see "Methods", Supplementary Fig. 8, Supplementary Data 4). There were 607 clusters that contained at least five homologs (Supplementary Fig. 8). Notably, 96 of these 607 gene clusters (15.8 %) were ribosomal proteins[23], which are known to be regulated translationally by feedback mechanisms.

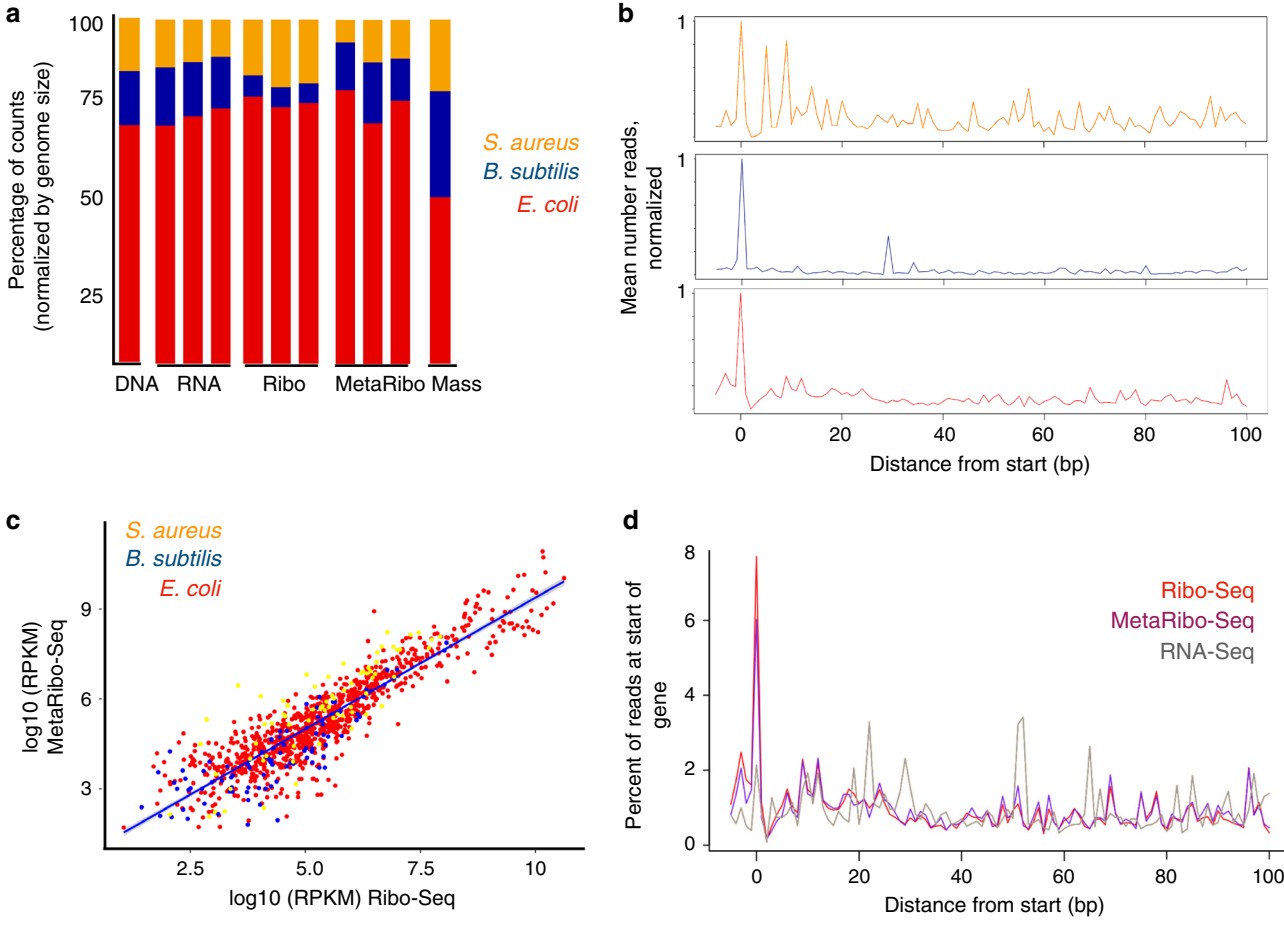

**Fig. 2 MetaRibo-Seq signal is consistent with signal observed in standard Ribo-Seq protocols. a** Proportion of count data from metagenomics, transcriptomics, MetaRibo-Seq, Ribo-Seq, and proteomics corresponding to *E. coli* (red), *B. subtilis* (blue), and *S. aureus* (yellow) in a three-member mock mixture. **b** MetaRibo-Seq signal at the start of genes for *E. coli*, *B. subtilis*, and *S. aureus*. **c** Scatterplot of MetaRibo-Seq read per kilobase, per million (RPKM) and Ribo-Seq RPKM log-scaled for the 1055 genes represented by all technologies in the mixture. Replicates reads were combined and treated as a single sample for this visualization. **d** MetaRibo-Seq, Ribo-Seq, and RNA-Seq signal at the start of genes in *E. coli*.

MetaRibo-Seq can also be used to measure translational regulation over time in the fecal microbiome. We performed RNA-sequencing on small mRNA fragments and MetaRibo-Seq on a second fecal sample (sample E2) collected 6 days after the initial collection of sample E from the same patient. In comparing gene expression in the *E. coli* strain between these two time points, 1,018 genes were significantly differentially transcribed, and 628 genes were significantly differentially translated (Supplementary Data 5, Supplementary Fig. 9). We found that 455 genes are regulated at a translational level, controlling for transcriptional changes (Supplementary Data 5, Supplementary Fig. 9).

**MetaRibo-Seq identifies thousands of small genes that are actively translated.** As small proteins are difficult to detect in metaproteomic experiments, one of our motivations in developing MetaRibo-Seq was to enable the detection of actively synthesized small proteins directly in fecal samples. Previous comparative genomic analyses on human associated metagenomes proposed 4,539 high confidence small gene families from a larger set of ~444,000 potential small gene families[4]. We first asked which of these 4,539 families (hereafter referred to as "4k families") could be supported by the MetaRibo-Seq data. We identified homologs of 1,337 of the 4539 families in the fecal metagenome samples A−E. Using an RPKM (reads per kilobase, per million) threshold of 10, we found that 623 of these 1,337

homologs (~47%) were synthesized into proteins (Fig. 5a, Supplementary Data 6).

One of the limitations of our previous efforts to identify small protein families was that we required a very high level of conservation across species and diverse representation across taxa (filtering out families with <8 different underlying sequences) in order to predict a family. Thus, our previous list[4] was enriched for true positives at the cost of a high false-negative rate. To overcome this limitation, we tested if MetaRibo-Seq could identify additional small gene families in the fecal microbiome. Using MetaRibo-Seq, we found evidence of translation for 2,091 additional small protein families (Fig. 5a, Supplementary Data 6), demonstrating utility of this method for small gene discovery. Because these 2,091 additional families were held to a relatively high RPKM threshold of 10, ~40% of these proteins contained a MetaRibo-Seq RPKM over tenfold higher than RNA-Seq RPKM (Supplementary Data 6). In our previous work, only 1,965 small proteins were predicted to be encoded in fecal microbiomes, based on an analysis of data from the Human Microbiome Project[24]. Thus, MetaRibo-Seq data more than doubled the number of small protein families predicted to exist in the fecal microbiome. We found that MetaRibo-Seq signal was enriched at the start codon compared to the gene body of these additional small gene families, as is expected for ribosome profiling across known genes (Fig. 5b). Relative to the 440,000 potential small gene families, these 2,091 small protein families were also

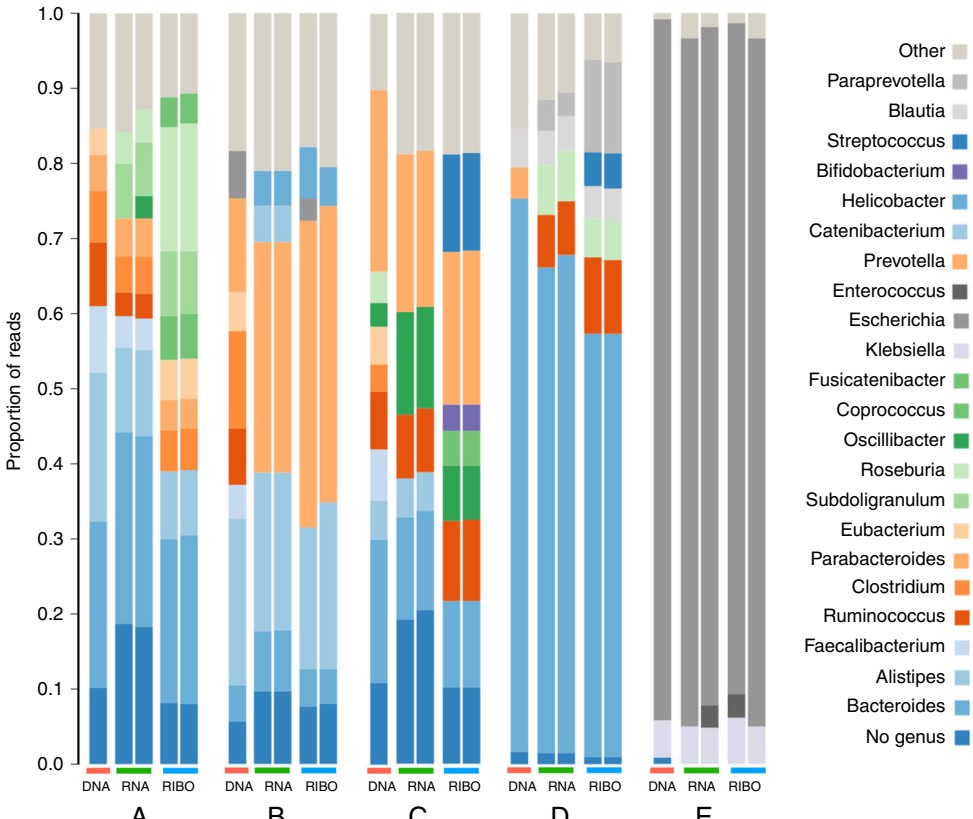

**Fig. 3 Taxonomic composition of samples across sequencing technologies.** Genus-level classifications of all sequencing technologies performed on samples A–E. Replicates for metatranscriptomics and MetaRibo-Seq are shown for reproducibility. Only classified reads are shown. Taxa represented below 3% relative abundance are grouped into "Other" for visual purposes.

enriched in known protein domains (~0.5% in the 440,000 versus ~6% in the 2,091 families, hypergeometric distribution test *p* value = 8.69e−90) (Fig. 5c).

Next, we tested if we could build a less stringent comparative genomics argument for these additional 2,091 small protein families using RNAcode[25], this time allowing for as little as two different DNA sequences per family (see "Methods"). Using a *p* value threshold of ≤0.05, 420 (20%) of these additional small protein families were also supported by comparative genomics (see "Methods", Fig. 5a). Upon comparing *E. coli* in sample E across two time points (E1 and E2), we found that 11 small protein families are translationally regulated, controlling for transcriptional changes (Supplementary Fig. 9, Supplementary Data 5). This demonstrated that these small protein families were likely translationally regulated in the fecal microbiome.

Although most of these 420 families were represented by a small number of homologs (Fig. 6a), some families were represented by a relatively large number of homologs (Fig. 6a). For example, family 29768 was a small protein family that escaped detection in our previous analysis due to the small number of unique DNA sequences; only six unique DNA sequences encoded the 337 different instances of this gene (Fig. 6b). Interestingly, 93% (314/337) of these homologs within family 29768 were coded for by a single unique DNA sequence (Fig. 6b). This conserved family was restricted to the genus *Bacteroides* but was found in 23 different species (Supplementary Data 6, 7). In 115/337 of the cases, family 29768 was encoded in the vicinity of a two-component system (Supplementary Data 6), suggesting that this small protein plays a role in signal transduction.

While highly conserved DNA sequences may be falsely rejected in predictions based on comparative genomic analyses, protein families that were undergoing rapid evolution may separate into multiple smaller clusters, and hence were less likely to appear in the previously predicted 4,539 set. Within the 420 small protein family set, 31 of the 60 families with known domains were assigned to the quorum sensing domain, *AgrD*, documented to be rapidly evolving[26]. Families 10356, 33628, 61327, and 7588 were collectively an example of four distinct families in the 420 set that share sequence homology with each other but were too divergent to be clustered together at 50% amino acid identity (Fig. 6c). Their genomic localization next to phage genes suggested that they were encoded on a prophage (Fig. 6c). The classification of the underlying contigs to diverse *Clostridia* clades (Supplementary Data 6, 7) suggested that this rapidly evolving protein was common to phages that infect *Clostridia*.

Unlike our previous comparative genomics approach, our current approach was able to identify additional small proteins that were less prevalent, as well as those that were encoded in less prevalent microbes. For example, family 333520 was a predicted transmembrane protein that is unique to *Prevotella* (Supplementary Data 6, 7), a relatively rare constituent of westernized gut microbiomes[24]. It shared sequence homology with family 357465, which was also a predicted transmembrane protein exclusive to *Prevotella* (Supplementary Data 6, 7).

Finally, families 386917 (55 members) and 386898 (68 members) were notable as they were the most highly translated proteins in our data, with RPKM values ranging between 5,090 and 310,648 (Supplementary Data 6). These families were part of the 2,091 additional small protein families predicted in this study,

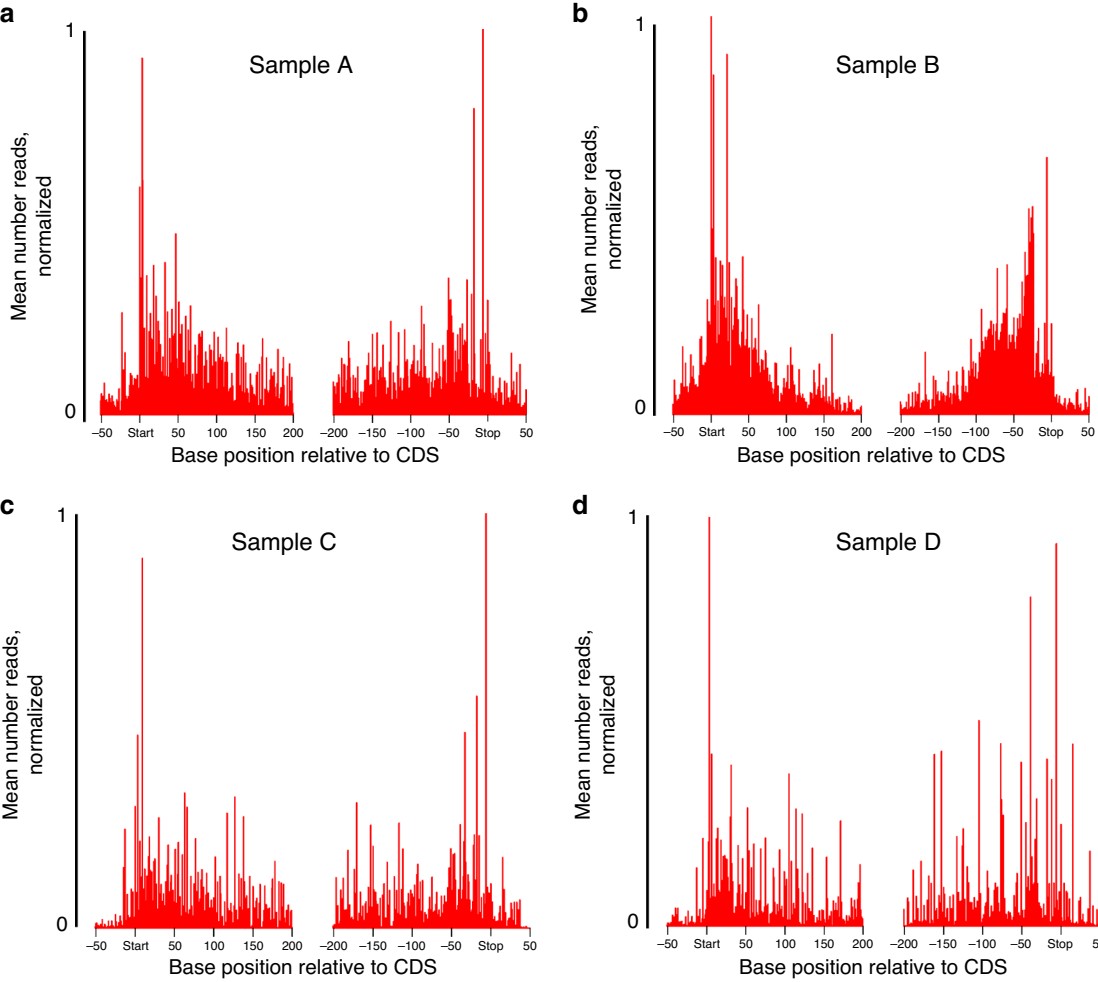

**Fig. 4 MetaRibo-Seq signal in fecal samples shows expected characteristics of ribosome profiling. a–d** Average MetaRibo-Seq signal across genes and flanking regions for Samples A−D, respectively. Every predicted open reading frame containing at least ten reads is included in the analysis. Replicates reads are combined and treated as a single read for this visualization.

but not the 420 set that was supported by a comparative genomics approach. We displayed the genomic context of the most highly translated homolog of family 386917 (MetaRibo-Seq RPKM = 310,648; Fig. 6d). These two Clostridiales-specific families (386898 and 386917) both encoded proteins that were 29 amino acids in length (Supplementary Data 6); family 386917 was predicted to be secreted. Even though we could not support these families with comparative genomics using RNAcode[25], their reproducible and exceptionally high translation rate suggested that these small proteins were genuine protein-coding genes.

## Discussion

To date, it has been challenging to comprehensively study fecal bacterial communities, or any complex system of bacteria, at the level of protein synthesis. Ribo-Seq of isolated and cultured bacterial strains has provided an understanding of the dynamic regulation of translation[15] and gene prediction, and has been especially useful in annotating small genes. However, the standard Ribo-Seq protocol cannot be applied to microbiome samples due to the high purity and cellular input required. Beyond bacteria, ribosome profiling has also been used to reveal the translational landscape and discover microproteins in the human heart[27]. Here, we developed MetaRibo-Seq to enable studies of gene translation within a natural microbiome setting in a way that is not restricted to culturable species. In addition, we demonstrate

that MetaRibo-Seq is a valuable method for detecting and validating small proteins, an area of increasing interest. Using MetaRibo-Seq, we validate 623 previously predicted small protein families[4] and also reveal thousands of additional small protein families in the fecal microbiome.

In the validation experiments we performed on mock communities, we observed a stronger correlation between MetaRibo-Seq signal and protein abundance compared to signal from RNA-Seq on small fragmented RNA and protein abundance. We did not formally compare standard RNA-Seq with longer RNA fragments to MetaRibo-Seq signal; thus, we cannot conclude that MetaRibo-Seq will always be a better predictor of protein abundance than RNA-Seq approaches. Our inability to observe a consistent improvement in MetaRibo-Seq correlation to proteomics in more complex fecal samples may be the consequence of equivalence of MetaRibo-Seq and RNA-Seq on small fragmented RNA as surrogate measures of protein level. Alternatively, this may be explained by the fact that only highly abundant proteins are detected by proteomics—and the genes encoding these highly abundant and thus detect proteins may not be subject to translational regulation. Future experiments that include standard RNA-Seq and that provide more comprehensive proteomic characterization of complex samples may enable a more robust comparison of MetaRibo-Seq vs. RNA-Seq as a surrogate for high-resolution, deep proteomics. Given that MetaRibo-Seq is a closer biological surrogate of protein synthesis

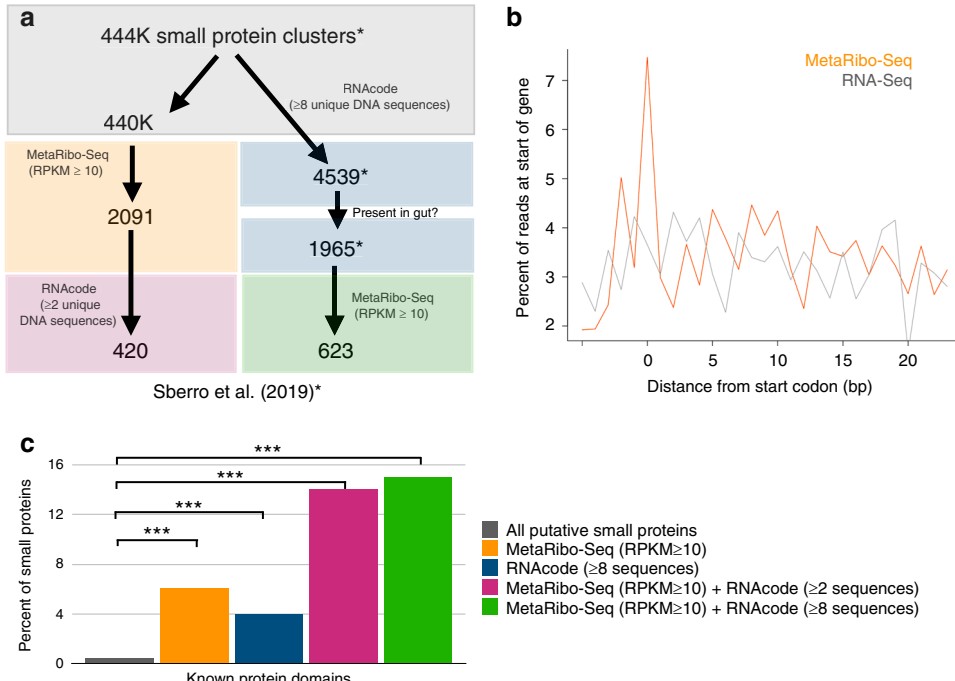

**Fig. 5 Protein synthesis of small protein families in the gut microbiome. a** Small gene families in HMPI-II metagenomes were identified, as performed previously[4]. Small proteins were predicted using MetaProdigal[42] after setting a 5 amino acid minimum for gene prediction. CD-Hit[55] clustered these small gene predictions across HMPI-II at 50% amino acid identity to create ~444,000 small gene clusters (see "Methods"). With comparative genomics using RNAcode[25], clusters with at least eight different sequences at a nucleotide level were further considered. ~4k small gene families were assigned significant *p* values with RNAcode[25], suggesting coding potential. 1,965 of these small gene families were found in the fecal samples. 623 of these gene families are synthesized into proteins in the fecal samples. An additional 2,091 small protein families were predicted based on MetaRibo-Seq signal. Among these, 420 small protein families were supported by comparative genomics using RNAcode[25], requiring at least two different sequences at a nucleotide level (see "Methods"). **b** Signal distribution of RNA-Seq and MetaRibo-Seq reads at and directly 3′ of the start codon for all 2,091 predicted small gene families. **c** Percentage of known protein domains among groups of small protein families. We compare our higher confidence sets of small proteins to the ~444,000 putative small protein clusters using the hypergeometric distribution and find that MetaRibo-Seq-predicted small proteins are significantly enriched for known protein domains compared to the initial, unfiltered ~444,000 putative small protein predictions. Comparing these ~444,000 protein clusters to clusters supported by MetaRibo-Seq, RNA-code (≥8 sequences), MetaRibo-Seq + RNA-code (≥2 sequences), and MetaRibo-Seq + RNAcode (8 sequences), we calculate *p* values of $3.87 \times 10^{-109}$, $8.05 \times 10^{-90}$, $8.07 \times 10^{-65}$, $5.38 \times 10^{-104}$, respectively. Significance was assigned as \*\*\**p* value < 0.001.

and the fact that the method is comparable in time and cost to standard RNA-Seq, it is possible that MetaRibo-Seq may be preferable to standard RNA-Seq when studying the coding moiety of the metagenome.

Beyond being a potential surrogate measure of protein levels, one uniquely valuable feature of Ribo-Seq experiments is the ability to test for differences in the abundance of transcripts alone vs. those that are being translated. MetaRibo-Seq answers similar questions but at a larger scale, and in a natural setting. This is interesting because in bacteria, many genes are regulated at the translational level. For example, genes involved in the process of translation itself are known to be translationally regulated via feedback mechanisms[23,28–31]. Most bacteria have not yet been studied at the translational level—and MetaRibo-Seq allows a view into translation of tens to hundreds of organisms simultaneously. For example, in our experiments, we found 96 clusters of genes encoding ribosomal proteins that are among the most different in terms of MetaRibo-Seq vs. RNA-Seq signal. This pattern is conserved across many different taxa and in organisms where Ribo-Seq has never been performed. This finding demonstrates that certain types of genes in microbiomes are being translated at different levels than transcribed in a reproducible and generalizable manner, and suggests that there may be common underlying mechanisms that regulate this phenomenon. Beyond the study of these housekeeping-type genes, we anticipate

that MetaRibo-Seq may provide valuable insights in studying regulation that occurs through post-transcriptional mechanisms like riboswitches, which have fascinating activities such as induction of the translation of antibiotic resistance genes[32]. Taken together, MetaRibo-Seq enables the measurement of post-transcriptional activities directly in microbiomes and may deepen our understanding of the relevance of these activities in clinically relevant phenotypes.

Finally, one of the major and perhaps most obvious applications of MetaRibo-Seq is in identifying coding regions, especially small coding regions, in microbiomes[18]. In previous work from our lab[4], we predicted 4,539 small gene families but were unable to validate the majority of them using evidence of translation of these genes. Because proteomic and Ribo-Seq support is a common way to validate predictions in the field of small protein research, we sought to apply MetaRibo-Seq to help validate some of our previous in silico predictions[33,34]. This resulted in validation of 623 of the previously proposed small gene families[4] and the prediction of an additional 2,091 small protein families that could not be detected using our comparative genomics approach. Though this work already more than doubles the number of small proteins predicted to exist in the human gut, MetaRibo-Seq will likely identify thousands more upon application to additional samples. In the future, it will likely prove useful in refining gene boundaries, especially with further benchmarking using other antibiotics.

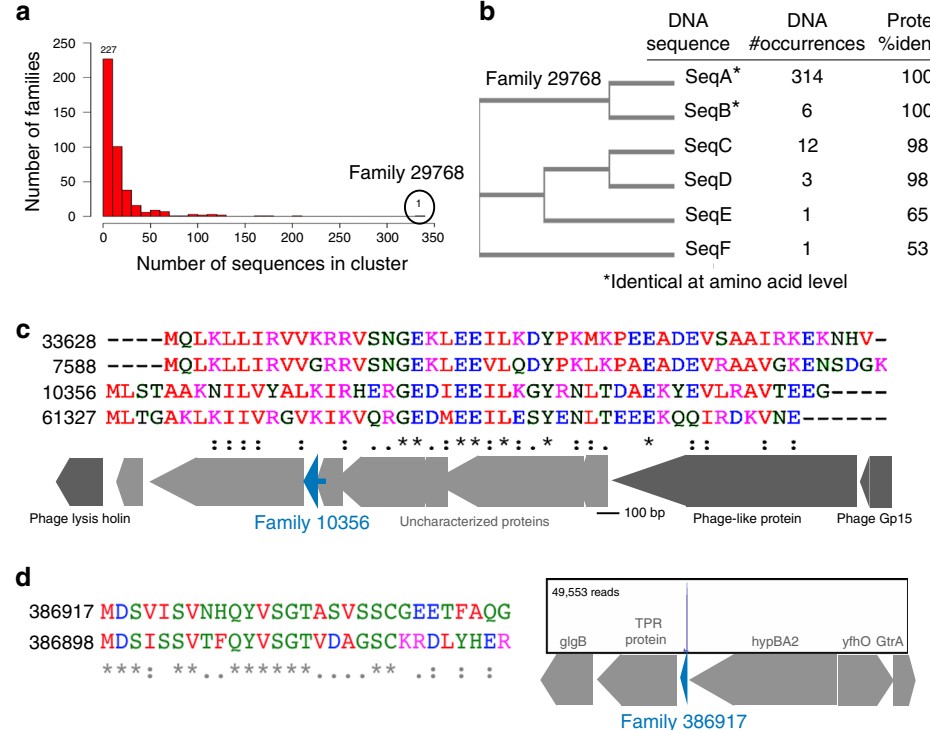

**Fig. 6 Characterization of small protein families predicted using MetaRibo-Seq. a** Histogram depicting the number of sequences representing each of the small protein families in the 420 set. **b** Cladogram representing the six unique DNA sequences that comprise family 29768, the family with the most homologs within the 420 set. We show the number of occurrence of each DNA sequence found within the family. We additionally show the amino acid percent identity, relative to SeqA, of the proteins encoded for by these sequences. **c** Multiple sequence alignment of four homologous small protein families 33628, 7588, 10356, and 61327 from the 420 set. Only representative protein sequences for each of these protein families are shown in the alignment. These four small protein families are often found near phage genes, as we depict using an example of the genomic neighborhood of family 10356. This specific genomic neighborhood is found in *Hungatella hathewayi* with the small protein (Genebank: CUP46342) represented as a blue arrow. **d** Multiple sequence alignment of two small protein families (386917 and 386898) within the 2,091 set (but not the 420 set), representing the most highly translated small proteins in our data. IGV visualization (Sashimi plot) of one 386917 homolog in sample D is shown.

Despite the utility of MetaRibo-Seq in measuring protein synthesis in microbiomes and in detecting small protein-coding genes, there are several limitations to this method. While MetaRibo-Seq signal does correlate with protein abundances, it does not perform as well as standard Ribo-Seq in terms of correlation to protein abundance in the mock community. Thus, in simple culturable communities, Ribo-Seq would be favored. MetaRibo-Seq also does not include steps to degrade RNAs with secondary structure, and this results in retention of structured RNAs. Of note, structured RNA contamination is a common issue in ribosome profiling protocols[7,11]. Though targeted approaches for specific bacteria have been successful in depleting tRNAs[26], an untargeted approach, which would be necessary here, has yet to be implemented in the literature. As this method is used and adapted, additional experimental modifications will likely help to address these limitations.

Overall, we show that translation can be comprehensively studied in microbiomes in a culture-free manner and that this method can shed light on translational regulation of genes. MetaRibo-Seq allows us to study protein synthesis across many bacterial taxa in feces at an unprecedented level of resolution and scale. By illuminating this facet of post-transcriptional regulation, we anticipate that future applications of MetaRibo-Seq will result in the discovery of thousands of additional small protein families and enable high-throughput study of how processes ranging from stress response to antibiotic resistance are regulated at the translational level.

## Methods

**Mock community culturing**. NR-2653 *E. coli* K-12 MG1655, NR-607 *B. subtilis* 168, and NR-45946 *S. aureus* RN4220 were obtained from BEI Resources. *Bacteroides thetaiotaomicron* VPI 5482 was obtained from ATCC (ATCC 29148). *E. coli*, *B. subtilis*, and *S. aureus* were grown individually in Luria-Bertani (LB) broth to an OD600 of 0.4 at 37 °C. Equal volumes of the bacteria were mixed thoroughly to create the three-member mock community. Metagenomics, metatranscriptomics, Ribo-Seq, MetaRibo-Seq, and proteomics were performed on this mixture. A second mock community was also created in which *E. coli* and *B. thetaiotaomicron* were grown anaerobically, both individually in Brain Heart Infusion (BHI) broth to an OD600 of 0.5 at 37 °C. Equal volumes of these bacteria were mixed to create a two-member mock community. Metatranscriptomics, MetaRibo-Seq, and proteomics were performed on this mixture.

**Mock community metagenomics**. Aliquots (25 mL) of the two mock communities were centrifuged in 50 mL tubes at 4000 × *g* at room temperature for 30 min. DNA was extracted from cellular pellets with DNA Stool Mini Kit (Qiagen) using the manufacturer's protocols. Samples were then exposed to bead beating for 3 min at room temperature. One nanogram of DNA was used to create Nextera XT libraries according to the manufacturer's instructions (Illumina).

**Mock community MetaRibo-Seq**. Aliquots (50 mL) of the community were centrifuged in 50 mL tubes at 4000 × *g* at room temperature for 30 min. Cell pellets were resuspended in 700 µL of RNAlater and stored at −80 °C for 1 week. These cells (150 mg) were suspended in 600 µL Qiagen RLT lysis buffer supplemented with 1% beta-mercaptoethanol, 0.3 U/µL Superase-In (Invitrogen), and 1.55 mM of chloramphenicol. This mixture was incubated at room temperature for 5 min. The suspension was subjected to bead beating for 3 min using 1.0 mm Zirconia/Silica beads. This was performed with a MiniBeadBeater-16, Model 607. The lysed solution was centrifuged at room temperature for 3 min at 21,000 × *g* to pellet cellular debris, and the supernatant was extracted to 2 mL tubes. The lysis supernatant was subjected to ethanol precipitation with 0.1% volume of 3 M sodium acetate and 2.5 M volumes of 100% ethanol. To precipitate, samples were incubated at −80 °C for 30 min, then centrifuged at 21,000 × *g* for 30 min at 4 °C. The pellet

of RNA and RNA-protein complexes was resuspended in MNase buffer. The buffer contained 25 mM Tris pH 8.0, 25 mM NH$_4$Cl, 10 mM MgOAc, and 1.55 mM chloramphenicol. One microliter of solution was diluted 20-fold and quantified with Qubit dsDNA HS Assay Kit (Invitrogen). MNase reaction mix was prepared as described[20], except this was scaled down to an input of 80 µg of RNA and 1 µL of NEB MNase 500 U/µL in a total reaction volume of 200 µL. The MNase reaction was incubated at room temperature for 2 h. All following steps were performed identically[20], except the tRNA removal steps were excluded. Briefly, 500 mL of polysome binding buffer was used to wash the Sephacryl S400 MicroSpin columns (GE Healthcare Life Sciences) three times—spinning the column for 3 min at 4 °C at 600 r.p.m. Polysome binding buffer consisted of 100 µL Igepal CA-630, 500 µL magnesium chloride at 1 M, 500 µL egtazic acid (EGTA) at 0.5 M, 500 µL of NaCl at 5 M, 500 µL Tris-HCl pH 8.0. at 1 M, and 7.9 mL of RNase-free water. The MNase reaction was applied to the column and centrifuged for 5 min at 4 °C. The flow-through was purified further with miRNAeasy Mini Kit (Qiagen) using the manufacturer's protocols. Elution was performed at 15 µL volume. rRNA was depleted using RiboZero-rRNA Removal Kit for Bacteria (Illumina) using the manufacturer's protocol, except all reaction volumes and amounts were reduced by 50%. This was purified with RNAeasy MinElute Cleanup Kit (Qiagen), eluting in 20 µL of water. The reaction, in 18 µL volume, was subjected to T4 PNK Reaction (NEB M0201S) with the addition of 1 µL Superase-In (Invitrogen), 2.2 µL 10× T4 PNK Buffer, and 1 µL T4 PNK (10 U/µL). This reaction was purified again with RNAeasy MinElute Cleanup (Qiagen). The concentration was determined with Qubit RNA HS Assay Kit (Illumina). With 100 ng of RNA as input, libraries were prepared using NEBNext Small RNA Library Prep Set for Illumina (NEB, E7330), using the manufacturer's protocols. DNA was purified using Minelute PCR Purification Kit (Qiagen). Libraries were sequenced with 1 × 75 bp reads on a NextSeq 500.

**Mock community Ribo-Seq.** Before harvesting mock community 1, it was treated with 0.1 mg of chloramphenicol per mL of culture. After 2 min, 50 mL aliquots of the community were centrifuged in 50 mL tubes at 4000 × g at room temperature for 30 min. Cell pellets were resuspended in 500 µL Ribo-Seq lysis buffer[20] (25 mM Tris pH 8.0, 25 mM NH$_4$Cl, 10 mM MgOAc, 0.8% Triton X-100, 100 U/mL RNase-free DNase I, 0.3 U/µL Superase-In, 1.55 mM Chloramphenicol, and 17 µM 5′-guanylyl imidodiphosphate). Lysis was performed using bead beating for 3 min in this lysis buffer. Twenty-five A260 units of RNA, measured using Nanodrop 2000, were treated with 6000U of MNase for 2 h at room temperature using MNase buffer to dilute as necessary. Five hundred milliliters of polysome binding buffer (100 µL Igepal CA-630, 500 µL magnesium chloride at 1 M, 500 µL EGTA at 0.5 M, 500 µL of NaCl at 5 M, 500 µL Tris-HCl pH 8.0. at 1 M, and 7.9 mL of RNase-free water) was used to wash a Sephacryl S400 MicroSpin column (GE Healthcare Life Sciences) three times—spinning the column for 3 min at 4 °C at 600 × g. The MNase reaction was applied to the column and centrifuged for 5 min at 4 °C. The flow-through was collected and was then purified further with miRNAeasy Mini Kit (Qiagen) according to the manufacturer's protocols, and the final sample was eluted from the miRNAeasy column in a volume of 15 µL in water. The sample was then taken forward for rRNA depletion using the MICROBExpress™ Bacterial mRNA Enrichment Kit (Invitrogen) according to the manufacturer's protocols. This reaction was purified with RNAeasy MinElute Cleanup Kit (Qiagen) using the manufacturer's protocols, eluting in 20 µL of water. The reaction, in 18 µL volume, was subjected to T4 PNK Reaction (NEB M0201S) with the addition of 1 µL Superase-In (Invitrogen), 2.2 µL 10× T4 PNK Buffer, and 1 µL T4 PNK (10 U/µL) for 1 h at 37 °C. This reaction was purified again with RNAeasy MinElute Cleanup (Qiagen) according to the manufacturer's protocols and the final sample was eluted in 10 µL of water. The final concentration of RNA was determined with Qubit RNA HS Assay Kit (Illumina). With 100 ng of RNA as input, libraries were prepared using NEBNext Small RNA Library Prep for Illumina (NEB, E7330), according to the manufacturer's protocols. DNA libraries were purified using Minelute PCR Purification Kit (Qiagen) using the manufacturer's protocols. Libraries were sequenced with 1 × 75 bp reads on a NextSeq 550.

**Mock community metatranscriptomics.** Aliquots (50 mL) of the community were centrifuged in 50 mL tubes at 4000 × g for 30 min at room temperature. Cell pellets were resuspended in RNA-Seq lysis buffer (25 mM Tris pH 8.0, 25 mM NH$_4$Cl, 10 mM MgOAc, 0.8% Triton X-100, 100 U/mL RNase-free DNase I, and 0.3 U/µL Superase-In). Lysis was performed using bead beating for 3 min in this lysis buffer. The mixture was centrifuged at 21,000 × g for 3 min at room temperature and the supernatant was collected. An equal volume of Phenol/Chloroform/Isoamyl Alcohol 25:24:1 (pH. 5.2) was applied and the sample was vortexed for 3 min. The mixture was centrifuged at 21,000 × g for 3 min at room temperature. The aqueous phase was extracted. This Phenol/Chloroform/Isoamyl Alcohol step was repeated once more. The final aqueous phase was ethanol precipitated using 2.5 volumes ethanol and 0.1 volumes sodium acetate. The resulting pellet was resuspended in 100 µL of water. The RNA was further purified using the RNAeasy Mini plus Kit (Qiagen) according to the manufacturer's protocols. Any remaining DNA was degraded via Baseline-ZERO-DNase (Epicentre) according to the manufacturer's protocols. RNA was fragmented for 15 min at 70 °C using RNA Fragmentation Reagent (Ambion) according to the manufacturer's protocols. At this point, the MetaRibo-Seq and small metatranscriptomics protocol completely converge. The

fragmented RNA was purified with miRNAeasy Mini Kit (Qiagen) according to the manufacturer's protocols and rRNA was eluted in a final volume of 15 µL of water. The resultant RNA was taken forward for rRNA depletion using the MICROBExpress™ Bacterial mRNA Enrichment Kit (Invitrogen), which was used according to the manufacturer's protocols. The resultant rRNA-depleted RNA was purified with an RNAeasy MinElute Cleanup Kit (Qiagen), eluting in 20 µL of water. The resulting RNA fragments, in 18 µL volume, were subjected to T4 PNK Reaction (NEB M0201S) with the addition of 1 µL Superase-In (Invitrogen), 2.2 µL 10× T4 PNK Buffer, and 1 µL T4 PNK (10U/µL) for 1 h at 37 °C. This reaction was purified again with RNAeasy MinElute Cleanup (Qiagen) according to the manufacturer's protocols. The final concentration of purified RNA was determined with Qubit RNA HS Assay Kit (Invitrogen). With 100 ng as input, libraries were prepared using NEBNext Small RNA Library Prep Set for Illumina (NEB, E7330), using the manufacturer's protocols. DNA was purified using MinElute PCR Purification Kit (Qiagen) according to the manufacturer's protocols. Libraries were sequenced with 1 × 75 bp reads on a NextSeq 500.

**Mock community metaproteomics.** Aliquots of the community (50 mL) were centrifuged in 50 mL tubes at 4000 × g at room temperature for 30 min. The cell pellet was resuspended in 2% SDS, 100 mM dithiothreitol (DTT), and 20 mM Tris HCl, pH 8.8 with protease inhibitor. These cells were subjected to bead beating for 3 min. The samples were then centrifuged for 3 min and the clarified lysate supernatant was collected. Lysate was prepared using Filter aided Sample Preparation (FASP)[35] with the same minor modifications previously documented[22]. Every following step involved a centrifugation step for 15 min at 14,000 × g. Protein concentrations were measured using Nanodrop 2000. Samples were diluted tenfold in 8 M urea and loaded into Microcon Ultracel YM-30 filtration devices (Millipore). They were washed in 8 M urea, reduced for 30 min in 10 mM DTT, and alkylated with 50 mM iodoacetamide for 20 min. Samples were washed three times in 8 M urea and two times in 50 mM ammonium bicarbonate. Trypsin (Pierce 90057) (1:100 enzyme-to-protein ratio) was added and incubated overnight at 37 °C. Into a new collection tube, samples were centrifuged and further eluded in 50 µL of 70% acetonitrile and 1% formic acid. The mixture was brought to dryness for 1 h using a Savant SPD121P SpeedVac concentrator at 30 °C, then resuspended in 0.2% formic acid[22].

**Metaproteomics.** These methods apply to all metaproteomics performed in this work (including mock communities and fecal communities). LC-MS/MS analysis was performed by the Stanford University Mass Spectrometry Facility using the Thermo Orbitrap Fusion Tribrid. A Thermo Scientific Orbitrap Fusion coupled to a nanoAcquity UPLC system (Waters, M Class) was used to collect mass spectra (MS). Samples were loaded on a 25-cm sub 100-µm C18 reverse phase column packed in-house with a 80-min gradient at a flow rate of 0.45 µL/min. The mobile phase consisted of: A (water containing 0.2% formic acid) and B (acetonitrile containing 0.2% formic acid). A linear gradient elution program was used: 0–45 min, 6–20% (B); 45–60 min, 35% (B); 60–70 min, 45% (B); 70–71 min, 70% (B); 71–77 min, 95% (B); 77–80 min, 2% (B). Ions were generated using electrospray ionization in positive mode at 1.6 kV. MS/MS spectra were obtained using collision-induced fragmentation (CID) at a setting of 35 of arbitrary energy. Ions were selected for MS/MS in a data-dependent, top 15 format with a 30-s exclusion time. Scan range was set to 400–1500 m/z. Typical orbitrap mass accuracy was below 2 p.p.m., for analysis. A 12-p.p.m. window was allowed for precursor ions and 0.4 Da for the fragment ions for CID fragmentation detected in the ion trap. Prokka-predicted[36] proteins were used as a reference database for protein detection using the Byonic proteomics search pipeline v 2.10.5 [37]. Byonic parameters include: spectrum-level FDR auto, digest cutter C-terminal cutter, peptide termini semi-specific, maximum number of missed cleavages 2, fragmentation type CID low energy, precursor tolerance 12.0 p.p.m., fragment tolerance 0.4 p.p.m., protein FDR cutoff 1%. These methods were performed by the Stanford Mass Spectrometry Facility (SUMS). Using spectral count output, normalized spectral abundance factor was calculated by in-house scripts.

**Subject recruitment.** MetaRibo-Seq was performed on fecal samples from individuals from a variety of health states. Informed consent was obtained for all participants. None of the participants received bacterial translation inhibitors. All subjects were recruited at Stanford University as a part of one of three IRB-approved protocols for tissue biobanking and clinical metadata collection (PIs: Dr. Ami Bhatt, Dr. Victor Henderson, Dr. David Miklos).

**Fecal sample storage.** Stool was immediately stored in 2 mL cryovials and frozen at −80 °C. Stool was not thawed until lysis. For RNA extraction applications, 1.3 g of fecal samples were preserved in 700 µL of RNALater (Ambion) at −80 °C.

**Cell lysis for Ribo-Seq, metatranscriptomics, and MetaRibo-Seq.** Stool (150 mg) was suspended in 600 µL Qiagen RLT lysis buffer supplemented with 1% beta-mercaptoethanol and 0.3 U/µL Superase-In (Invitrogen). For MetaRibo-Seq lysis, 1.55 mM of chloramphenicol was also added to this lysis solution, and the solution was incubated at room temperature for 5 min. The suspension was subjected to bead beating for 3 min using 1.0 mm Zirconia/Silica beads. This was performed

with a MiniBeadBeater-16, Model 607. The lysed solution was centrifuged at room temperature for 3 min at $21,000 \times g$ to pellet cellular debris, and the supernatant was extracted to 2 mL tubes.

**Metagenomics**. DNA was extracted from fecal samples with DNA Stool Mini Kit (Qiagen) using the manufacturer's protocols. Samples were exposed to bead beating for 3 min. One nanogram of DNA was used to create Nextera XT libraries according to the manufacturer's instructions (Illumina). Metagenomic libraries were sequenced with $2 \times 101$ bp reads on an Illumina HiSeq 4000 instrument.

**MetaRibo-Seq**. The lysis supernatant was subjected to ethanol precipitation with 0.1% volume of 3 M sodium acetate and 2.5 M volumes of 100% ethanol. To precipitate, samples were incubated at $-80\,°C$ for 30 min, then centrifuged at $21,000 \times g$ for 30 min at $4\,°C$. This was a rough purification specifically implemented to enable suspension of concentrated RNA from reasonable input of fecal sample. The pellet of RNA and RNA−protein complexes was resuspended in MNase buffer. The buffer contained 25 mM Tris pH 8.0, 25 mM NH₄Cl, 10 mM MgOAc, and 1.55 mM chloramphenicol. To resuspend, we quickly broke the pellet apart with a pipette tip and vortexed for 15 s. One microliter of solution was diluted 20-fold and quantified with Qubit dsDNA HS Assay Kit (Invitrogen). MNase reaction mix was prepared as described[20], except this was scaled down to an input of 80 µg of RNA and 1 µL of NEB MNase 500 U/µL in a total reaction volume of 200 µL. The MNase reaction was incubated at room temperature for 2 h. All following steps were performed identically[20], except the tRNA removal steps were excluded. Briefly, 500 mL of polysome binding buffer was used to wash the Sephacryl S400 MicroSpin columns (GE Healthcare Life Sciences) three times— spinning the column for 3 min at $4\,°C$ at 600 r.p.m. Polysome binding buffer consisted of 100 µL Igepal CA-630, 500 µL magnesium chloride at 1 M, 500 µL EGTA at 0.5 M, 500 µL of NaCl at 5 M, 500 µL Tris-HCl pH 8.0. at 1 M, and 7.9 mL of RNase-free water. The MNase reaction was applied to the column and centrifuged for 5 min at $4\,°C$. The flow-through was purified further with the miRNAeasy Mini Kit (Qiagen) using the manufacturer's protocols. Elution was performed at 15 µL volume of water. rRNA was depleted using RiboZero-rRNA Removal Kit for Bacteria (Illumina) using the manufacturer's protocol, except all reaction volumes and amounts were reduced by 50%. This was purified with RNAeasy MinElute Cleanup Kit (Qiagen), eluting in 20 µL of water. The reaction, in 18 µL volume, was subjected to T4 PNK Reaction (NEB M0201S) with the addition of 1 µL Superase-In (Invitrogen), 2.2 µL 10× T4 PNK Buffer, and 1 µL T4 PNK (10U/µL). This reaction was purified again with RNAeasy MinElute Cleanup (Qiagen). The concentration was determined with Qubit RNA HS Assay Kit (Illumina). With 100 ng as input, libraries were prepared using NEBNext Small RNA Library Prep Set for Illumina (NEB, E7330), using the manufacturer's protocols. DNA was purified using Minelute PCR Purification Kit (Qiagen). Libraries were sequenced with $1 \times 75$ bp reads on a NextSeq 500.

**Small metatranscriptomics of fecal samples**. We performed metatranscriptomics as follows: 15 µL of proteinase K (Ambion, 20 mg/mL) was added to 600 µL of lysate. After incubation for 10 min at room temperature, samples were centrifuged at $21,000 \times g$ for 3 min and the supernatant was collected. An equal volume of Phenol/Chloroform/Isoamyl Alcohol 25:24:1 (pH. 5.2) was applied and vortexed for 3 min. The mixture was centrifuged at $21,000 \times g$ for 3 min. The aqueous phase was extracted. This phenol chloroform step was repeated once more and the aqueous phase was extracted. This final aqueous phase was ethanol precipitated with 0.1% volume of 3 M sodium acetate and 2.5 M volumes of 100% ethanol. The resulting pellet was resuspended in 100 µL of water. The RNA was further purified using the RNAeasy Mini plus Kit (Qiagen) using the manufacturer's protocols. Any remaining DNA was degraded via Baseline-ZERO-DNase (Epicentre) using the manufacturer's protocols. RNA was fragmented for 15 min at $70\,°C$ using RNA Fragmentation Reagent (Ambion) using the manufacturer's protocols. At this point, the MetaRibo-Seq and small metatranscriptomics protocol completely converge. The fragmented RNA was purified with the miRNAeasy Mini Kit (Qiagen) using the manufacturer's protocols. Elution was performed at 15 µL of water. rRNA was depleted using RiboZero-rRNA Removal Kit for Bacteria (Illumina) using half reactions of the manufacturer's protocol. This was purified with the RNAeasy MinElute Cleanup Kit (Qiagen), eluting in 20 µL of water. The fragments, in 18 µL volume, were subjected to T4 PNK Reaction (NEB M0201S) with the addition of 1 µL Superase-In (Invitrogen), 2.2 µL 10× T4 PNK Buffer, and 1 µL T4 PNK (10 U/µL). This reaction was purified again with RNAeasy MinElute Cleanup (Qiagen) using the manufacturer's protocols. The concentration was determined with Qubit RNA HS Assay Kit (Invitrogen). With 100 ng as input, libraries were prepared using NEBNext Small RNA Library Prep Set for Illumina (NEB, E7330), using the manufacturer's protocols. DNA was purified using the MinElute PCR Purification Kit (Qiagen) using the manufacturer's protocols. Libraries were sequenced with $1 \times 75$ bp reads on a NextSeq 500.

**Differential centrifugation and FASP for metaproteomics**. To remove human proteins, fecal samples were subjected to differential centrifugation. One hundred milligrams of fecal sample was suspended in 1× phosphate-buffered saline (PBS) in 1.7 mL Eppendorf tubes. The tubes were centrifuged at $600 \times g$ for 1 minute at

room temperature. The supernatant was collected in a clean Eppendorf tube and centrifuged at $10,000 \times g$ for 1 minute at room temperature. The supernatant was decanted and the pellet was resuspended in 1 mL of PBS. The process was repeated once more. The final pellet was resuspended in 2% SDS, 100 mM DTT, and 20 mM Tris HCl, pH 8.8 with protease inhibitor. These cells were subjected to bead beating for 3 min with a MiniBeadBeater-16, Model 607. Zirconia/silica beads (1 mM) were used. Tubes were centrifuged for 3 min and clarified lysate in the supernatant was collected. Lysate was prepared using FASP[35] with the same minor modifications previously documented[22]. Every step involved a centrifugation step for 15 min at $14,000 \times g$. Samples were diluted tenfold in 8 M urea and loaded into Microcon Ultracel YM-30 filtration devices (Millipore). They were washed in 8 M urea, reduced for 30 min in 10 mM DTT, and alkylated in 50 mM iodoacetamide for 20 min. Samples were washed three times in 8 M urea and two times in 50 mM ammonium bicarbonate. Trypsin (Pierce 90057) (1:100 enzyme-to-protein ratio) was added and incubated overnight at $37\,°C$. Into a new collection tube, samples were centrifuged and further eluded in 50 µL of 70% acetonitrile and 1% formic acid. The mixture was brought to dryness for 1 h using a Savant SPD121P SpeedVac concentrator at $30\,°C$, then resuspended in 0.2% formic acid[22].

**De novo assembly**. Quality-trimmed metagenomic reads were assembled using metaSPAdes 3.7.0[38]. For all samples, a maximum of 60 million metagenomic reads were used to generate assemblies. Samples sequenced to higher depth were randomly subsetted to 60 million for assembly purposes to both ensure relatively similar numbers of gene predictions and limit computational requirements in assembly and downstream predictions.

**Read mapping, gene prediction and annotation**. Reads were trimmed with trim galore version 0.4.0 using cutadapt 1.8.1[39] with flags –q 30 and –illumina. Reads were mapped to the annotated assembly using bowtie version 1.1.1[40]. To avoid all possible conservation conflicts in downstream differential analysis, only perfect, unique short read alignments were considered. IGV[41] was used to visualize coverage. Prokka v1.12[36] was used to predict genes from the metagenomics assemblies using the –meta option. Annotations were facilitated by many dependencies[42–45]. For small protein predictions, Prodigal[42] was performed after lowering the size threshold from 90 bases to 15 bases.

**Read density as a function of position**. MetaRibo-Seq reads were mapped to their metagenomic assemblies. The assembly and aligned reads were analyzed with RiboSeqR[46]. Ribosome profiling counts for predicted coding sequences (CDSs) were determined with the sliceCounts function. CDSs were filtered to contain at least ten reads.

**Taxonomic classification of technologies**. Reads mapping specifically to Prokka-predicted[36] coding regions were counted. We classified every predicted gene in these metagenomes using One Codex[47]. We determined the classification of reads based on the classification of the gene it mapped to. This enabled fair comparisons between technologies, as the small metatranscriptomics and MetaRibo-Seq reads can be too small to classify individually with $k$-mer-based approaches. Though metagenomic reads were long enough to be classified directly, they were also subject to the same analysis. Thus, all taxonomy plots represent entire gene classifications and are dependent on the assembly.

**Differential analysis**. The number of reads mapping to a given region was calculated with BEDtools multicov version 2.27.1[48]. Strandedness was enforced for metatranscriptomics and MetaRibo-Seq. All differential analyses were performed using these counts with all conditions performed in duplicate via DESeq2[49]. A gene was considered differential if it had log2fold change above 1 or below −1, while also reaching an FDR < 0.05. Results were reported as tables. In the case of translational regulation (sample E compared to sample E2), we modified the model to control for RNA levels (design = ~samplegroups + samplegroups:type,). Heatmaps were created using gplots[50]. Reads per kilobase million calculations were performed using in-house scripts.

**Statistical analysis**. All Pearson correlations were calculated in R using the Hmisc package[51]. Scatterplots were created with ggplot2[52]. For all scatterplots and histograms shown, replicates reads were combined and treated as a single sample. Significance between Pearson's $r$ was assigned via cocor[53]. Significant differences between RPKM values were assigned using the Kolmogorov−Smirnov test. Significance was assigned as $*p$ value < 0.05, $***p$ value < 0.001. Zou's[54] 95% confidence intervals were considered significant (assigned as $***$) if there is no overlap with 0 in the interval.

**Protein clustering analysis for MetaRibo-Seq vs. transcriptomics**. For analyses independent of gene annotation, proteins that were translated at different levels than transcribed, discussed in the differential analyses methods, were clustered using Cd-hit[55] with 70% amino acid identity. Representative sequences were input into Blast2GO[56] using the nr database.

**Triplet periodicity analysis**. Using the same default parameters as read density as a function of position, triplet periodicity was called using RiboSeqR[46]. To analyze triplet periodicity of specific genera, assembled contigs were classified using One Codex[47]. Contigs that classified into a specific genus were binned together. Only reads mapping specifically to these bins were considered.

**Clustering of small proteins from HMP-I-II**. Contigs from the 1773 HMPI-II metagenomes containing at least 5 Mbp of total contig sequence were downloaded from https://www.hmpdacc.org/hmasm2. MetaProdigal[42], using a cutoff length of 15 bp, was used to predict genes. Small ORFs, encoding potential proteins 5−50 amino acids including start and stop codons, were considered. These small proteins were clustered using CD-Hit[55] with parameters: -n 2, -p 1, -c 0.5, -d 200, -M 50000, -l 5, -s 0.95, -aL 0.95, -g 1. This resulted in 444,054 clusters, which were identical to those previously generated.

**Identifying homologs of the ~444,054 clusters in samples A−E**. For samples A−E, MetaProdigal[42], using a cutoff length of 15 bp, was used to predict genes. Small ORFs, encoding potential proteins 5−50 amino acids including start and stop codons, were considered. Small proteins predicted in samples A−E were queried against representatives of each of the ~444,000 clusters, using BLASTp[43] with word-size of 2. Hits were considered significant if: $e$ value ≤ 0.05 and the length of the hit was between 90 and 110% of the length of the small protein.

**Demonstrating protein synthesis of small gene families**. For each sample (A−E), we considered all predicted genes, including these small genes. The total number of MetaRibo-Seq reads mapping to all of these genes was calculated. As previously described in the "Methods" section, bowtie 1.1.1[40] was used to map reads. The number of reads mapping to a given region was calculated with BEDtools multicov version 2.27.1[48]. Strandedness was enforced. We calculated RPKM for all of these genes for each sample (A−E). If a given small protein demonstrates translation (MetaRibo-Seq RPKM > 10) and is homologous to one of the ~444,000 potential small gene families, we considered this evidence of protein synthesis of these small gene families.

**Small protein statistical analysis**. To test for enrichment in proportion of predictions with protein domains across our assigned confidence levels, hypergeometric distribution tests were performed.

**Assessment of homology between small protein families**. Using BLASTp[43], we blasted the 2091 small protein families containing homologs with MetaRibo-Seq signal (RPKM > 10) against the initial 4000 gene families proposed previously. We defined rapid evolution as instances in which any homolog of the 2091 small protein clusters significantly hit ($e$ value ≤ 0.05) representative protein sequences for the initially proposed 4000 small gene families.

**Taxonomic classification of small protein families**. Contigs containing any homolog of the 2091 small protein families were classified using Kraken2 v2.0.8[57] with a custom database constructed from RefSeq[58] and GenBank[59]. To visualize the classifications within each small protein family, Krona[60] was utilized.

**Genomic neighborhood analysis of small protein families**. MetaProdigal[42] was used to annotate genes on contigs containing any homologs of the 2091 small protein families. Amino acid sequences of genes that are at maximum ten genes away from the small protein along these contigs were searched against the Conserved Domain Database (CDD)[61], using RPS-blast[43]. A hit was considered significant if: $e$ value ≤ 0.05 and the protein aligns to at least 80% of the PSSM's length.

**Cellular localization of small protein families**. This was performed on all proteins within the 2091 small protein families proposed. To predict if these proteins are secreted, SignalP-5.0[62] was run with default parameters both with "gram+" and "gram−". To predict if these proteins are transmembrane, TMHMM[63] was run on the same set of proteins with default parameters. A small protein family was considered transmembrane/secreted if ≥80% of the members were predicted to be such.

**Antimicrobial peptide identification**. AmPEP[64] was applied (default parameters) on representatives of the 2091 small protein families.

**Guidelines for extraction of all contigs associated with a specific family of interest**. We provide these small protein families at a DNA and amino acid level in Supplementary Data 2. If you would like to extract the contigs these regions are predicted from, please follow instructions previously presented under "Guidelines for extraction of all contigs associated with a specific family of interest"[4]. Additionally, we provide krona plots for each family (2091 .html files in total) in which all contigs for the family were taxonomically classified and can be interactively viewed.

**Reporting summary**. Further information on research design is available in the Nature Research Reporting Summary linked to this article.

## Data availability

Data generated in this study are available on Sequencing Read Archive (SRA) and can be downloaded under the bioproject accession PRJNA510123. Data underlying Figs. 2a and 5c and Supplementary Figs. 1b, 1c, 1d, 2a, 2b, 2d, 2f, 3a, 4c, 4d, 4e, 4f, and 8a are provided as Source Data files. Other data are available from the corresponding author upon reasonable request.

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

## Acknowledgements

The authors would like to thank Tessa M. Andermann, Ekaterina Tkachenko, and Joyce B. Kang for major contributions to the fecal biobank of blood and marrow transplant patients at Stanford hospital, as well as Christina Wyss-Coray for her leadership in overseeing the Alzheimer's sample collection used in this study. We would like to thank the patients and nurses involved in collection. Victor Henderson and Tony Wyss-Coray provided valuable feedback with respect to sample collection strategies. We thank Max (Mahmoud) Al-Bassam for helpful discussions, and Anshul Kundaje and Georgi Marinov for helpful computational analysis guidance. We would also like to thank Stanford University Mass Spectrometry for performing and analyzing mass spectrometry on the FASP fecal samples and Michael Bassik, Gaelen Hess, and Roarke Kamber, for allowing us to access their NextSeq 550 system. We thank Soumaya Zlitni for guidance in generating figures. Sequencing costs were supported via NIH S10 Shared Instrumentation Grant (1S10OD02014101) and Damon Runyon Clinical Investigator Award to A.S.B., Stanford ADRC grant # P50AG047366. B.J.F. is supported by the National Science Foundation Graduate Research Fellowship DGE-114747.

## Author contributions

The project was conceptually designed by B.J.F and A.S.B.. Experiments were performed by B.J.F.. Data analysis for method benchmarking purposes was analyzed by B.J.F.. Data analysis for small protein predictions was analyzed by B.J.F and H.S.. Data were interpreted by B.J.F., H.S. and A.S.B.. Figures were created by B.J.F.. The paper was prepared by B.J.F. and A.S.B.. All authors discussed the results and implications and provided feedback on the manuscript.

## Competing interests

The authors declare no competing interests.
