## [Peer Review File · Nature Communications]

Reviewer #2 (Remarks to the Author):

The authors have strengthened the case for MetaRiboSeq being able to identify novel sORFs, which was my major concern with the resubmitted Nature Biotechnology manuscript. However, I am still not completely convinced that the method works well for this application. The current method is very simple: predicted ORFs need to have MetaRiboSeq RPKM >10 to be declared real. Since RPKM is determined in part by RNA abundance, this approach is overly simple. It would be better to require both a minimum level of RPKM and a minimum ratio of coverage in MetaRiboSeq relative to RNA-seq data. I also think the method can be benchmarked more effectively. The enrichment of MetaRiboSeq signal in ORFs vs UTRs is reassuring, as is the enrichment at start codons. If the authors ignore the E. coli genome annotation, does their approach identify known E. coli ORFs from the mock community data? If so, how accurate are the start codon calls? There are lots of known sORFs in E. coli, including many that were recently identified (PMID 30837344).

I would also be interested to see how the metagene plots showing MetaRiboSeq coverage look if the set of genes analyzed is limited to those that are not close to a gene on the same strand, e.g. for three genes in the following orientation, $\leftarrow \rightarrow \leftarrow$, the middle gene start and stop codons are not close to a gene on the same strand, so it could be analyzed for MetaRiboSeq coverage at both the start and end of the gene; similarly, the middle gene in $\rightarrow \rightarrow \leftarrow$ could be analyzed MetaRiboSeq coverage at the end but not the start of the gene. By disregarding genes that have a nearby ORF on the same strand, it should be easier to tell how selective the method is for ORFs versus UTRs. These plots would also be good to see as heatmaps, so coverage across all genes can be visualized rather than an average.

We thank the reviewers for their helpful comments. We have detailed our responses below. The reviewers comments are in blue, and our responses are in black.

The authors have strengthened the case for MetaRiboSeq being able to identify novel sORFs, which was my major concern with the resubmitted Nature Biotechnology manuscript.

We greatly appreciate the feedback this reviewer has provided over this and the previous rounds of review. These comments have enabled us to substantially strengthen our report of the very first method for Ribo-Seq profiling of microbiome samples and our subsequent application of MetaRibo-Seq to sORF predictions.

However, I am still not completely convinced that the method works well for this application. The current method is very simple: predicted ORFs need to have MetaRiboSeq RPKM >10 to be declared real. Since RPKM is determined in part by RNA abundance, this approach is overly simple. It would be better to require both a minimum level of RPKM and a minimum ratio of coverage in MetaRiboSeq relative to RNA-seq data.

Response 1.1

We agree that this is a simple method to determine if a sORF is translated, which we based on an EBI standard. The Expression Atlas (EMBL-EBI) suggests that genes between 0.5 and 10 RPKM are lowly expressed while those between 10 and 1000 are at a medium expression level. Thus, we selected 10 RPKM as the lower threshold for the “medium expression level” to balance sensitivity and specificity. The reviewer is also correct that RNA abundance (and also genomic abundance) can affect the MetaRibo-Seq RPKM of a sORF. For example, if a sORF transcript is found at low levels in a fecal sample (which could be because it is not highly expressed in the organism or because the organism itself is of low abundance in the sample), the sORF on that transcript is more likely to be lowly translated. This means that even highly translated proteins in low abundance organisms may not reach an RPKM > 10, resulting in a high false negative rate. This is a general issue even in metagenomics - organisms and genes can be too low in abundance to detect.

Given these inherent challenges, we sincerely appreciate the reviewers help to address this limitation. We have performed the analysis the reviewer suggested. In summary, we calculated the ratios of Ribo-Seq RPKM to RNA-Seq RPKM. Approximately 40 percent of the newly called small proteins have a MetaRibo-Seq RPKM over 10-fold higher than the corresponding RNA-Seq RPKM. Thus, many of the small proteins with Ribo-Seq RPKMs greater than 10 also have high translation efficiencies. This occurs primarily because our RPKM cutoff is relatively high for MetaRibo-Seq and there are likely many sORFs transcribed at low levels in the samples that are being predicted with MetaRibo-Seq because they are translated at high efficiency.

While we have performed this analysis and now report these values (lines 186-188), we caution that one limitation of using this ratio to inform predictions is that a gene does not necessarily need to be translated more than transcribed to be a bona fide protein-encoding gene. If the

RNA-Seq value is much higher than the Ribo-Seq value, it suggests a low translation efficiency but translation nonetheless. Our approach clearly has a potentially high false negative 'rate' as we chose a relatively high RPKM cutoff at 10, while an argument can be made that 0.5 (from EMBL-EBI) is enough to suggest that a gene is at least expressed at a low level. This more stringent RPKM cutoff of >10, though simple, will identify sORF transcripts that are relatively highly translated - either because the organism is abundant, the gene is highly transcribed, or is transcribed at at least a minimum level and is translated at a very high rate.

I also think the method can be benchmarked more effectively. The enrichment of MetaRiboSeq signal in ORFs vs UTRs is reassuring, as is the enrichment at start codons. If the authors ignore the *E. coli* genome annotation, does their approach identify known *E. coli* ORFs from the mock community data? If so, how accurate are the start codon calls? There are lots of known sORFs in *E. coli*, including many that were recently identified (PMID 30837344).

Response 1.2

We thank the reviewer for these suggestions to improve benchmarking of MetaRibo-Seq. The reviewer alludes to the fact that Ribo-Seq has been used effectively and successfully in bacterial isolate experiments and eukaryotic experiments to perform de novo gene prediction. We agree that MetaRibo-Seq may be employed, after careful continued development and benchmarking, for more extensive gene prediction without requiring a priori ORF identification.

As a reminder, in this manuscript, in contrast to the entirely de novo approaches employed in PMID 30837344 (the manuscript the reviewer alluded to), we do not use MetaRibo-Seq alone to identify gene boundaries. Rather, we first use candidate sORF annotations reported in a previous manuscript from our lab (Sberro et al, Cell 2019; PMID 31402174). These potential sORFs are a "superset" from which we then use MetaRibo-Seq to assess if these predicted sORFs are transcribed and translated. As the work in this manuscript directly followed a very large effort in our lab to bioinformatically improve high-confidence predictions of sORFs (Sberro et al, Cell, 2019), we had chosen to leverage this helpful "superset" of data as a starting point for validating sORFs.

Here, the reviewer suggests an entirely alternative and valid approach for calling sORF boundaries that is entirely 'de novo', not requiring an a priori candidate gene to be identified. We thank the reviewer for suggesting that we perform an analysis to determine if we can identify known *E. coli* ORFs from the mock community data in a way similar to what was described in PMID 30837344. We have followed this suggestion and have provided the analysis below. Of note, in carrying out this analysis, we did experience one limitation in performing the identical analysis to that which was described in this manuscript: the list of 160,955 potential sORFs that were considered in this manuscript was not publicly accessible. To overcome this limitation, we used the prodigal pipeline to predict putative sORF (predicted 90 sORFs). Reassuringly, we found that the known small proteins (predicted in PMID 30837344 and reported in Table 1 of that manuscript) from *E. coli* within our mock community were more likely to be translated than

potential small proteins predicted using Prodigal. Thus we find that MetaRibo-Seq benchmarks well against this alternative approach, as well.

With respect to the reviewer's thoughtful question about identifying start codons and identifying gene boundaries with Ribo-Seq, we agree that this is an excellent application of MetaRibo-Seq. As the reviewer suggests, Ribo-Seq can be used to identify gene boundaries, given high enough Ribo-Seq coverage across an ORF. An ideal way to achieve this is to modify the Ribo-Seq protocol by adding specific antibiotics to the fresh bacterial samples to stall ribosomes at the start codon. This modified Ribo-Seq protocol was used in PMID 30837344. As this was the first effort to develop a method for applying Ribo-Seq on metagenomic samples, we have not benchmarked these specialized types of modifications in this first paper on MetaRibo-Seq. We are keen to do so in the future, but do note at least 2 likely limitations we/others may face in trying to do so: (i) This would require immediate mixing of ribosome-stalling antibiotics into a fecal sample prior to freezing; (ii) Many gut microbiome bacteria are resistant to ribosome-stalling antibiotics - and thus, this approach may not be truly generalizable to all microbes in the microbiome. That said, we fully agree that this is an exciting potential future experiment that we would be interested to perform. Accordingly, we have modified the discussion section to highlight these exciting potential future applications (lines 294-306).

Finally, we appreciate the reviewer's impression that MetaRibo-Seq is a reliable protocol and is capable of so much more than we are presenting in this initial manuscript. We are keen to carry out these additional experiments, but agree that to do so properly would require extensive experiments and proper benchmarking. As this manuscript already introduces the very first protocol that enables ribosomal profiling in a mixed microbial/microbiome sample, we feel that these additional experiments, while very exciting and promising, are out of the scope of this piece of work. We agree that we have not fully pushed the limits of MetaRibo-Seq's capabilities and that future experiments may enable us and others to comprehensively predict sORFs and to define gene boundaries. Both of these applications would be optimally addressed by making experimental modifications (such as introducing different antibiotic treatments just after sample collection) to sample processing. We are reassured and excited that the reviewer feels that this method holds promise for these future applications and we anticipate that we and others will be exploring these opportunities in the near future.

I would also be interested to see how the metagene plots showing MetaRiboSeq coverage look if the set of genes analyzed is limited to those that are not close to a gene on the same strand, e.g. for three genes in the following orientation, $\leftarrow \rightarrow \leftarrow$, the middle gene start and stop codons are not close to a gene on the same strand, so it could be analyzed for MetaRiboSeq coverage at both the start and end of the gene; similarly, the middle gene in $\rightarrow \rightarrow \leftarrow$ could be analyzed for MetaRiboSeq coverage at the end but not the start of the gene. By disregarding genes that have a nearby ORF on the same strand, it should be easier to tell how selective the method is for ORFs versus UTRs. These plots would also be good to see as heatmaps, so coverage across all genes can be visualized rather than an average.

Response 1.3

We appreciate this creative approach to help determine whether MetaRibo-Seq signal is enriched in genes vs. UTRs in the fecal samples. In our previous response to reviewers' comments, we included the fold-enrichment of RPKM values of known genes in *E. coli* compared to known corresponding UTR regions using RNA-Seq, Ribo-Seq, and MetaRibo-Seq. These results showed that MetaRibo-Seq signal is indeed substantially enriched in ORFs vs. UTRs in a well-controlled setting. To highlight this analysis, which demonstrates enriched signal in ORFs compared to UTRs, we have now added this analysis to the manuscript (Figure S1C). We performed this in *E. coli* due to several limitations with the non-model organisms that are present in mixed microbiome samples. Specifically, in non-model organisms, it is difficult to make the distinction between UTRs and untranscribed regions, as orthogonal methods have not been performed on these organisms to inform UTR boundaries.

Figure S1C: Histogram showing the fold enrichment of signal across coding regions compared to signal across UTR regions for RNA-Seq, MetaRibo-Seq, and Ribo-Seq in *E. coli*. Replicates were used to calculate standard deviation.

Focusing only on genes with no other nearby genes on the same strand, as the reviewer suggests, is a way to ensure no signal from other genes gets included in the analysis. However, there are some limitations that may affect our confidence in such results. First, our annotations of the genomes of these organisms may be incorrect - if this is the case, we may incorrectly conclude that no gene is present in a region where there is, indeed, a gene. Second, we do not know much about UTRs in non-model organisms. Thus, it would still be difficult to determine which of these genes contain UTRs and how long each UTR would be. Thus, in summary, we lack the resolution to confidently compare specific genes to their specific UTRs in fecal samples and we fear that we'd have to make many relatively arbitrary and not well founded assumptions to carry out this analysis. It is certainly an exciting exploratory analysis, but something that we feel would be out of the scope of the current work, given that the manuscript is already very dense and long.

That said, we have already addressed this issue in a slightly different way with our metagene plots, as the reviewer points out. Specifically, we have been able to show is strong average enrichment in coding relative to surrounding regions. In future work and as we gain a better understanding of UTR boundaries in more organisms, this resolution will likely become attainable.